# Regulatory diversity in *Bacillus thuringiensis cry* genes reveals flexible evolutionary strategies for *in vivo* toxin expression

Isabel Gómez☯, Blanca I. Garcia-Gómez☯, Nathaly A. do Nascimento☯, Oscar Infante, Pablo Emiliano Cantón, Sabino Pacheco, Angel E. Peláez-Aguilar, Jorge Sanchez, Mario Soberón, Alejandra Bravo *

Departamento de Microbiología Molecular, Instituto de Biotecnología, Universidad Nacional Autónoma de México (UNAM), Cuernavaca, Morelos, Mexico

☯ These authors participate equally
* alejandra.bravo@ibt.unam.mx

## Abstract

*Bacillus thuringiensis* (Bt) is a highly effective insect pathogen, largely due to the expression of diverse insecticidal proteins upon sporulation. Among them, the three-domain Cry protein family represent the largest family, targeting a wide range of insect species and nematodes. While it is proposed that they have evolved from a common ancestral gene, the comparative analysis of their upstream regulatory regions revealed significant variability. To investigate this divergence, we employed **M**ultiple **E**xpectation maximization for **M**otif **E**licitation (MEME) and **F**ind **I**ndividual Motif **O**ccurrences (FIMO) motive discovery tools, to identify conserved regulatory elements, including canonical -10 and -35 promoter motifs and Shine-Dalgarno (SD) sequences. Our analyses clearly revealed that upstream regulatory sequences are not conserved across the entire *cry* family. However, we identify subsets of genes with similar insect specificity which shared conserved motif architectures in their upstream regulatory sequences, suggesting a correlation between regulatory evolution and host range. Conversely, some proteins targeting the same insect order (e.g., Cry1 and Cry9Ca or Cry3 and Cry8) showed to be regulated by entirely different upstream sequences, indicating that Bt has evolved multiple regulatory strategies to achieve similar expression patterns. To test relevance of the upstream sequences, we cloned *cry1Ab* and *cry4Ba* genes under the control of heterologous upstream regions: P1P2 from lepidopteran-specific *cry1Aa* gene, and P4 from dipteran-specific *cry4Ba* gene. These constructions were expressed in non-toxic Cry⁻ acrystalliferous Bt-backgrounds with distinct host specialization: Bt subsp *thuringiensis* 407 strain (lepidopteran adapted) and Bt subsp *israelensis* 4Q7 strain (dipteran adapted). Gene expression was assessed *in vitro* and *in vivo* after oral infection of lepidopteran and dipteran larvae with purified spores. Our findings indicate that Cry protein expression is influenced by both, the promoter identity, and Bt strain background, underscoring

**Data availability statement:** All relevant data are in the manuscript and its supporting information files.

**Funding:** This work was supported in part by Dirección general de Asuntos del personal Académico, Universidad Nacional Autónoma de México, (PAPPIT-UNAM IN200625 to AB), Consejo Nacional de Ciencia y Tecnología (CONAHCyT-Ciencia de Frontera CF2023-I-386 to AB), and Consejo Nacional de Ciencia y Tecnología (CIORGANISMOS-2025-56 to MS). The funders had no role in study design, data collection and analysis, decision to publish, or preparation of the manuscript.

**Competing interests:** The authors have declared that no competing interests exist.

the evolutionary and functional significance of upstream regulatory sequences in the diversification and ecological success of Bt.

## Author summary

*Bacillus thuringiensis* (Bt) is an effective insect pathogen that produces a wide variety of insecticidal proteins, including Cry toxins, which are highly specific toward certain lepidopteran, coleopteran, dipteran and nematodes. However, although Cry proteins share a common evolutionary origin, similar structural organization and mechanism of action, the evolution of their regulatory sequences revealed significant variability that is much less explored. Here, we systematically analyzed the upstream regions of *cry* genes, to understand how these regulatory elements evolved. We found that these regions are remarkably diverse, with partial conservation among toxins that target similar insect order, suggesting some adaptation to specific hosts during their evolution. Interestingly, we also identify cases where *cry* genes active against same insect order, such as *cry3* and *cry8* genes (both active against coleopteran), exhibit entirely distinct regulatory architecture, suggesting independent evolutionary paths, reflecting genetic flexibility, likely contributing to Bt´s ecological adaptation to different insect host. To explore their functional relevance, we engineered Bt strains to express the same Cry protein under different upstream sequences and evaluated toxin expression and activity in insect larvae. Our results suggest that Bt has evolved multiple transcriptional strategies to fine-tune *cry* gene expression, enabling successful adaptation to diverse insect environments.

## 1. Introduction

*Bacillus thuringiensis* (Bt) is one of the most successful insect pathogens, primarily due to its capacity to produce diverse insecticidal proteins [1]. These proteins have been widely used in pest control, either in specific formulations targeting mosquitoes as well as multiple lepidopteran and coleopteran crop pests, or through their expression in transgenic crops such as Bt-corn, Bt-cotton and Bt-soybean [2].

Bt produces different families of insecticidal proteins including Cry, Cyt, Vip, Tpp, App, Gpp, Vpa and Vpb [3]. More than 755 Cry members have been classified into 57 groups and 166 subgroups, based on their amino acid sequence identity [3]. Cry proteins sharing less than 45% amino acid identity are assigned different numbers (e.g., Cry1, Cry2), while subgroup with 45–75% identity are designated with different capital letter (e.g.,Cry1A, Cry1B). Variants with more than 75% but less than 95% identity are labeled with lowercase letters (e.g., Cry1Aa, Cry1Ab) and finally variant with more than 95% identity are distinguished with a different number. This classification broadly correlates with Cry proteins specific toxicity against larval stages of different insect orders such as Lepidoptera (butterflies and moths), Coleoptera (beetles and weevils), and Diptera (flies and mosquitoes) as well as nematodes [3].

Structural studies of Cry proteins revealed a conserved three-domain structure organization [4]. Domain I, a seven α-helix bundle, is involved in oligomerization and pore formation, disrupting midgut cells and killing larvae. Domains II and III conformed mainly by β-strands, participate in determining specificity by interacting with larval insect membrane midgut proteins. Specifically, Domain II forms a β-prism with exposed loops that bind receptors, while Domain III is a β-sandwich, where β-16 and β-22 strands are also involved in specificity [4].

Phylogenetic analysis suggests that Cry proteins evolved from a common ancestor, sharing conserved motifs in key structural regions, such as helix α-5 inside Domain I and strands β-17 and β-23 localized in Domain III, stabilizing the β-sandwich structure [5]. Two evolutionary processes involved in insect specificity have been proposed: independent evolution of Cry proteins and domain III swapping among different Cry members. These processes led to proteins with similar mode of action but distinct insect specificities [4,5].

In general, all Cry proteins are primarily produced during the Bt sporulation phase, forming parasporal inclusions that can account for up to 25% of the dry weight of sporulated cells. Sporulation is triggered by nutrient limitation, leading to the activation of key regulatory pathways. The master regulator, phosphorylated Spo0A, initiates sporulation by activating sigma factors $\sigma^H$ and $\sigma^E$ in the mother cell, and $\sigma^F$ in the forespore. During early sporulation ($t_5$), Sigma factor $\sigma^E$ is expressed in the mother cell and activates expression of *cry* genes, later ($t_{11}$) $\sigma^E$ activates $\sigma^K$, which further regulates Cry protein expression [6,7]. It was shown that during the transition-phase, the $\sigma^H$ may induce low level expression of some *cry* genes such as *cry1, cry4, cry8,* and *cry11* [6–9] and that the expression of several *cry* genes, including *cry1, cry4, cry8, cry11, cry18, cry64Ba,* and *cry64Ca,* is driven by $\sigma^E$ and $\sigma^K$, while *cry2* is exclusively regulated by $\sigma^E$ [6–11]. As an exception, *cry3A* expression is controlled by $\sigma^A$ during vegetative growth with enhanced expression during stationary phase by an unidentified regulator [12].

Beyond sigma factors, additional regulators may influence *cry* expression. Spo0A represses $\sigma^H$ dependent expression of *cry4* and *cry11,* via a conserved upstream Spo0A binding site, while it positively regulates *cry1Ac,* since Spo0A mutants showed lower expression of *cry1Ac* gene, probably due to the lower expression of the sporulation sigma factors [13,14]. SpoIIID positively affected expression of *cry4B* [15]. Pyruvate dehydrogenase E2 subunit binds an inverted repeat upstream of *cry1A* located 260 bp upstream from the *cry1A* promoter region, enhancing its expression, implying a connection with carbohydrate catabolism [16]. In *Bacillus subtilis*, pyruvate dehydrogenase participates in the expression of some sporulation genes [17]. Additionally, catabolite repression by glucose and fructose 6-phosphate downregulates *cry4A* via a CcpA-HPr complex which binds to a *cre* site overlapping the *cry4Aa* promoter [18].

These findings suggest that *cry* gene regulation is a complex process, involving more than just sigma factor participation. One question that remains not answered is how conserved is the regulation of Cry proteins among the entire *cry* family genes? Analysis of upstream regulatory sequences showed that they are highly variable across the *cry* gene family, yet this variability has not been extensively analyzed. Expression comparisons of *lacZ* fusions with *cry1Ac, cry3A, cry4A,* and *cry8E* upstream sequences showed differential expression levels under same *in vitro* conditions, supporting the idea that different upstream sequences show diverse efficiency [19].

Here, we analyzed the upstream regulatory sequences of the *cry* gene family and found a high variability. However, subsets of genes with similar insect host specificity, targeting Lepidoptera, Coleoptera, or Diptera, showed conserved organization of upstream motifs, suggesting that regulatory evolution has paralleled their host adaptation. Notably, important exceptions were also found, for example *cry3* and *cry8* genes (both active against coleopteran larvae) show completely distinct architecture of their regulatory motifs. These analyses indicated that, despite a common evolutionary origin of Cry proteins, their regulatory regions have followed distinct evolutionary paths. To analyze the functional consequences of these divergences, we conducted *in vivo* infection assays using purified spores, where the same *cry* gene was expressed under the control of different upstream sequences in different Bt bacterial backgrounds. Under these infection conditions, the Cry toxins are expected to be expressed inside the larval gut during infection. We observed differential toxicity, depending on the promoter used and the bacterial background. Together, these data suggest that Bt has developed

diverse strategies to express *cry* genes and has also evolved to ensure optimal *cry* gene expression during host infection, contributing to its broad ecological success.

## 2. Results

### 2.1. Analysis of *cry*-gene upstream sequences

To explore the conservation or variability of *cry* gene promoter regions, we retrieved upstream sequences of all *cry* genes reported in the Bacterial Pesticidal Protein Resource Center (BPPRC) [3]. We selected a representative *cry* gene from each subgroup of *cry* gene with different nomenclature, meaning different number, capital and lowercase letters (e.g., Cry1Aa, Cry1Ab etc.) since each of these subgroups share same upstream sequences, prioritizing sequences with the longest upstream region available in the GenBank database (S1 Table). Because these sequences showed very high divergence, and different length, traditional alignments and phylogenetic analyses were not feasible. For this reason, we employed a different strategy, using the algorithmic tool of Multiple Expectation maximizations for Motif Elicitation (MEME), which identifies conserved sequence motifs without gaps in a set of unaligned DNA sequences, thus global alignments were not required [20]. A total of 82 DNA upstream sequences (S1 Table) were analyzed with MEME to identify conserved motifs (S1 Fig). This analysis confirmed extensive variability across major *cry* gene groups. However, we observed that subsets of genes within certain *cry* groups (e.g., *cry1, cry1I, cry8* and *cry3* groups) shared conserved motif organization of MEME-blocks, suggesting a common evolutionary origin of those regulatory upstream regions. We hypothesized that these conserved motifs may correspond to functional *cis*-regulatory elements that influence gene expression.

For a more detailed comparison, we selected upstream regions of comparable length (specifically, 170–250 bp upstream of the ATG start codon). Exceptions were made for *cry4Aa* and *cry4Ba*, whose sequences were extended to 404 and 316 bp, respectively, to include their experimentally mapped promoter regions (S2 Table). MEME analysis of this subset of 50 sequences identified 24 distinct conserved motif-blocks, ranging from 14 to 75 bp (S3 Table). The sequence alignment of each of these blocks is shown in S3 Table, displaying their strong internal conservation. Fig 1 shows the distribution and arrangement of these blocks, color-coded and numbered as in S3 Table. In Fig 1, conserved motif architectures of MEME block-motifs were grouped into shaded rectangles, and the insect host specificity of each *cry* gene is indicated by the color of these rectangles: green for lepidopteran, brown for coleopteran, blue for dipteran and yellow for nematode specific sequences. Dual-specificity toxins (e.g., Lep/Col or Lep/Dip) are represented with combined colors. The *cry18Ca* sequence remained uncolored, as its insect specificity is unknown.

Among the lepidopteran-specific *cry* genes, four distinct upstream motif architectures were identified. The first group includes *cry1A, cry1C, cry1D, cry1E, cry1F, cry1G, cry1H* and *cry1J* genes, which shared a conserved core composed of motif-blocks 1–4, with some variants containing blocks 7 or 8. Within this group, *cry1Ca, cry1D*, *cry1G, cry1Hb* and cry*1Ka* were more divergent, lacking some blocks. Interestingly, *cry1Ca*, although primarily lepidopteran-active, has also been reported to show toxicity against mosquito larvae [21,22]. The second group of lepidopteran-specific *cry* sequences is formed by *cry2Ac* and *cry9Ca,* which were nearly identical (99% identity), composed of blocks 16, 12, 8 and 13. The third group includes *cry2Ab* and *cry2Ad* genes. They exhibit high similarity (91% identity) with a unique architecture including blocks 14, 15, and 11, unrelated to the first two groups. (Fig 1 and S3 Table). Notably, despite the high primary sequence similarity among Cry2 proteins, the upstream gene regions of *cry2Ab* and *cry2Ad* genes, diverged significantly from those of *cry2Aa* and *cry2Ac*, highlighting substantial regulatory diversification (Fig 1). These differences likely reflect their distinct evolutionary histories, since *cry2Aa* and *cry2Ac* genes are part of a three-gene operon in which the promoter is located upstream of *orf1* [23,24], whereas *cry2Ab* gene is located separately and does not form part of an operon arrangement [24]. Finally, the fourth regulatory configuration type was observed for *cry9Aa,* which contained only motif-block 18.

In the case of proteins that displayed dual toxicity (lepidopteran-coleopteran or lepidopteran-dipteran), we found that these sequences shared specific blocks with some of the lepidopteran sequences, reflecting hybrid regulatory

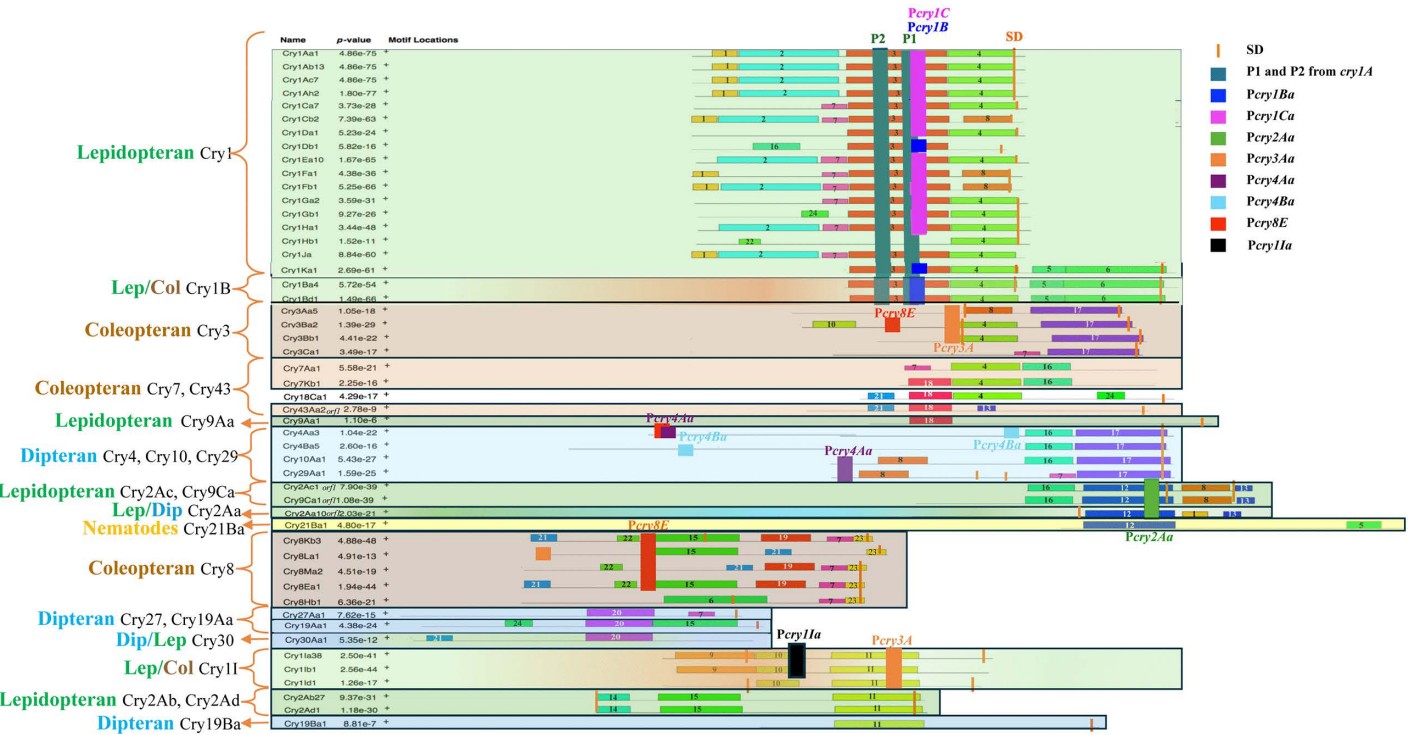

**Fig 1. Alignment of motif blocks identified by MEME in the upstream regions of 50 *cry* genes.** Upstream sequences (~250 bp) of 50 *cry* genes were analyzed using the "Multiple Expectation Maximization for Motif Elicitation" (MEME) algorithm, revealing 24 distinct conserved motif blocks (colored and numbered, as S3 Table). For *cry4Aa* and *cry4Ba*, the analyzed regions were extended to include their experimentally mapped promoter sequences. Previously described –10 and –35 promoter elements and putative Shine–Dalgarno (SD) sequences are also indicated. Shaded rectangles denote insect host specificity: green for lepidopteran, brown for coleopteran, blue for dipteran, and yellow for nematode. Dual-specificity toxins (e.g., Lep/Col or Lep/Dip) are represented by combined colors. The *p*-value associated with each motif represents the probability that a random sequence of equivalent composition and length would match the motif with an equal or higher log-likelihood ratio at a given position, under the MEME background model. Motifs with *p* < 0.001 were considered highly significant.

architectures. For example, *cry1Ba* and *cry1Bd* (Lep/Col specificity) [25] were closely related (94% identity) and shared motif-blocks 3 and 4 with the first group of lepidopteran sequences, but they also included blocks 5 and 6 at their 3' end, which were absent from all other *cry* sequences analyzed except *cry1Ka* (Fig 1). The upstream regions of *cry1B* and *cry1K* were highly conserved relative to each other (89–93% identity).

However, another dual Lep-Col group, the *cry1I* group of sequences [25,26], showed distinct architecture, composed by blocks 9, 10 and 11. Block 11 is shared with the lepidopteran-specific *cry2Ab* and *cry2Ad*. The Lep-Dip dual-specific *cry2Aa* [27,28], showed moderate similarity (56–61% identity) to the lepidopteran specific *cry2Ac* and *cry9Ca,* sharing blocks 12 and 13*.*

Among the dipteran-specific toxins, three upstream motif architectures emerged. The first group includes *cry4*, *cry10* and *cry29* genes. They shared block 17, while some sequences showed blocks 8 or 16 in conserved positions. The second group of dipteran-specific toxins includes *cry19Aa* and *cry27,* sharing block 20, which was also present in Lep/Dip dual-specific *cry30Aa.* The third type consists of *cry19Ba*, which uniquely contained block 11. Interestingly, *cry19Aa* and *cry19Ba* encoded highly similar proteins but showed divergent upstream organizations, again suggesting independent evolution of their regulatory regions.

The coleopteran-specific upstream sequences of *cry* genes also clustered into three groups. The first group has *cry3* genes, sharing moderate similarity (69–75% identity) including block 17. Block 10 of *cry3Ba* also appeared in the

upstream region of *cry1I* genes (Lep/Col dual specific). The second group consist of *cry8* genes which contained blocks 21, 22, 15, 19, 7 and 23, where blocks 19 and 23 were unique to this group. The third group contain *cry7* and *cry43Aa*. They displayed a conserved arrangement of blocks 18, 4 and 16. The upstream sequence of *cry18Ca* is related to this group.

Overall, these data indicate that Bt *cry* genes exhibit highly diverse upstream regulatory architectures. In some cases, conserved motifs arrangements were associated with shared insect specificity, suggesting co-evolution of regulatory sequence with host range. However, in other cases, closely related toxins display divergent upstream architectures, emphasizing the evolutionary flexibility of *Bt* in developing distinct transcriptional strategies.

## 2.2. Identification of known promoter elements among *cry* genes

To further assess how this regulatory diversity relates to transcriptional control, we next examined the presence of canonical promoter elements within the upstream regions of *cry* genes. We performed a motif-based analysis using FIMO tool [29], which systematically searches for specific known short motives such as previously characterized -10 and -35 promoter elements, Shine-Dalgarno (SD) sequences and putative protein-binding sites across the 82 selected upstream sequences listed in S1 Table. The FIMO results are summarized in S4 Table, with selected motifs mapped in Fig 1. Additionally, these regulatory elements are highlighted within their corresponding gene sequences in MEME block alignment shown in S3 Table.

Our analysis revealed a striking conservation of the overlapping P1P2 promoters arrangement, previously described in *cry1A* genes [7,12]. These elements were consistently located within MEME-block 3 in all *cry1* genes, except in the upstream sequence of *cry1I* genes that lack these promoters and also lack block 3 (Fig 1 and S3-S4 Tables), strongly suggesting a shared regulatory origin for the subset of *cry1* gene family containing block 3. Notably, the MEME-block 3 was absent from all other *cry* gene groups, highlighting the specificity of this conserved promoter structure.

Interestingly, promoter sequences previously mapped for *cry1Ba* and *cry1Ca* genes [10] were also located within MEME-block 3, showing 73.3% identity between their promoter sequence and 80–90% identity to the P1 promoter of *cry1Aa* gene, respectively. Homologous sequences were also detected in the upstream region of *cry1A, cry1B, cry1C, cry1D, cry1Ea, cry1F, cry1Gb, cry1Ha* and *cry1Ka,* reinforcing the existence of a conserved regulatory module within these genes (Fig 1 and S4 Table).

The promoter of *cry1Ia*, was reported to be located 119 bp upstream from the start codon and to be $\sigma^E$ dependent [30]. This putative promoter region was identified at the end of MEME-block 10 in both *cry1Ia* and *cry1Ib* sequences (Fig 1 and S4 Table). A similar sequence was also found in *cry29Aa,* although with lower identity (74%) and located further upstream (-528 bp from the ATG) (S4 Table). Block 10 is also present in *cry3Ba,* but the specific *cry1Ia* promoter motif was not found at the end of block 10 from *cry3Ba* (S3 Table).

For the *cry3Aa* gene, the first identified promoter was located at -129 bp [31].Similar sequences were found in *cry3Ba* and *cry3Bb* in 5`end from MEME-block 4, but not in *cry3Ca* (Fig 1 and S4 Table). Intriguingly, this promoter motif also appeared in coleopteran specific *cry8La* and in the dual Lep-Col specific *cry1I* genes (Fig 1 and S4 Table). In *cry1Fa,* a homologous sequence was detected at position -304 (S4 Table). Subsequent studies identified an alternative functional *cry3Aa* promoter at -558 bp [12,32]. FIMO analysis confirmed the presence of this promoter sequence in *cry3Ba*, but its identification in *cry3Bb* or *cry3Ca* genes was not possible due to sequence length limitations in the BPPRC database and the lack of available Bt genomes information carrying these genes.

The $\sigma^H$-dependent promoter of *cry8Ea* [9] was detected between MEME-blocks 22 and 15 (Fig 1 and S4 Table). Similar sequences were present in the same location of other *cry8* genes, except *cry8Hb* (Fig 1). Notably, *cry3Ba* also showed a sequence partially resembling the *cry8Ea* promoter region (66% identity) (S4 Table).

In relation to the promoter region described for the dipteran specific *cry4Aa* gene, that was described in position -364 bp upstream to the ATG codon [15], we detected homologous promoter sequences in *cry10Aa* (80% identity) and

*cry29Aa* (76.6% identity)*,* both encode dipteran-specific toxins (Fig 1 and S4 Table). Likewise, a homologous sequence to the *cry4Ba* promoter, originally mapped at -312 bp [15], was identified in the upstream region of *cry4Aa* at -180 (83.3% identity) and in *cry28Aa* (60% identity)(Fig 1 and S4 Table).

The *cry2Aa* and *cry2Ac* genes have been reported to be transcribed as part of a three-gene operon [23,24], in contrast to *cry2Ab,* which is located separately. Notably, no promoter has been identified for *cry2Ab*, and no expression has been detected, suggesting it may be a cryptic gene [24,33]. We found that the *cry2Aa* promoter [23,24], is located within MEME-block 12 (Fig 1 and S3 and S4 Tables), and a sequence with 84% identity was also present in the MEME block 12 region of *cry2Ac* and *cry9Ca* (Fig 1 and S3-S4 Tables). Remarkably, the upstream sequence of *cry2Ac* was 99% identical to that of *cry9Ca* (Fig 1 and S4 Table)*,* suggesting that they shared evolutionary origin. Both genes are organized in operons that include an identical *orf1* gene. The MEME-Block 12 was exclusively identified in *cry2Aa, cry2Ac*, *cry9Ca* and *cry21Ba*; however, the specific *cry2Aa* promoter sequence was absent in *cry21Ba* (Fig 1 and S3 and S4 Tables).

Finally, we examined promoter regions previously reported for additional *cry* genes not represented in Fig 1, such as *cry11Aa* [34], *cry18Aa* [35], *cry32Aa* [36], *cry40-cry34* [37], *cry41Ca* and *cry45Ba* [38] (S4 Table). These data were included in S4 Table to provide a broader comparative framework and are consistent with the promoter diversity observed across other *cry* gene families.

Overall, this comparative analysis revealed both conserved and lineage-specific promoter architectures among *cry* genes. The conserved P1P2 promoter module within MEME-block 3 of the *cry1* family represents a robust regulatory signature likely inherited from a common ancestor, while other promoter arrangements, such as the σ$^H$-dependent *cry8Ea* promoter, illustrate specialized adaptations. Similarly, the presence of homologous promoter elements in unrelated groups (e.g., *cry3* and *cry8*) points to possible functional convergence. Taken together, these findings highlight the coexistence of deeply conserved regulatory modules and lineage-specific innovations, reflecting the evolutionary flexibility of Bt in fine-tuning *cry* gene expression.

## 2.3. Presence of multiple Shine–Dalgarno (SD) elements and localization of other binding motifs among cry genes

FIMO was also used to identify putative SD sequences within *cry* gene upstream regions. Our analysis confirmed that 64% of the SD sequences found in *cry* genes were located approximately 11–15 bp upstream of the ATG start codon, a spacing consistent with their functional role as ribosome binding site (RBS) elements (Fig 1 and S4 Table). Interestingly, several upstream regions contained multiple SD sequences (S4 Table). Specifically, two SD sequences were identified in *cry1Ba, cry1Ia, cry1Ka, cry2Ab, cry2Ac, cry2Ad, cry3Aa, cry3Bb, cry8Hb, cry8Kb, cry9Aa, cry9Ca, cry19Aa* and *cry32A*. Moreover, some genes, including *cry1Id, cry2Aa, cry8La, cry10Aa, cry26Aa* and *cry29Aa* contained up to three SD sequences in their upstream regions (Fig 1 and S4 Table). The functional relevance of these additional SD sequences remains unclear. For *cry3Aa,* it has been proposed that the presence of an extra SD sequence facilitates binding of the ribosomal 30S subunit, thereby blocking ribonuclease access and extending mRNA half-life [31].

Consistent with this, the STAB-SD element, previously described as stabilizing *cry3Aa* mRNA [32] was also identified in the upstream regions of several *cry* genes by our FIMO analysis (S4 Table). Notably, in four cases (*cry3Aa, cry3Ca, cry9Aa* and *cry10Aa*), this motif was not found located in the proximity of the start codon. The participation of these SD boxes in *cry* gene regulation is not known and still requires further work.

FIMO analysis of additional motifs revealed that pyruvate dehydrogenase (PDH) binding sites, previously reported in *cry1Aa* [16,17], were present only in a subset of *cry1* sequences including *cry1Aa, cry1Ab, cry1Ac, cry1Ah, cry1Cb, cry1Ea, cry1Fb,* and *cry1Ha*, in similar location as the *cry1Aa* sequence. These motifs were absent in all other *cry* upstream regions analyzed.

Similarly, the Spo0A binding site identified upstream of *cry4Aa* [13] was only detected in *cry1Fb, cry2Ad*, and *cry31A*. The catabolite-responsive element (Cre) motif previously described for *cry4Aa* [18] was found exclusively in the upstream

regions of *cry1Cb*, *cry18Ca* and *cry43Ba* (S4 Table). Finally, recently the transcriptional activator VipR, that controls *vip3A* that was identified in *cry1Ia* gene [39]. We used that sequence to search with FIMO in the upstream sequences of *cry* genes. This VipR motif was found to be located 319 bp upstream of the *cry1I* start codon, and similar motifs were also detected upstream of *cry9Ca-orf1*, *cry2Ab*, and *cry9Aa* genes (S4 Table).

Taken together, these data indicate that additional regulatory motifs within *cry* gene upstream regions are not broadly conserved across the entire gene family, but instead occur only in specific subsets of *cry* genes.

### 2.4. *In vitro* expression of *cry1Ab* and *cry4Ba* genes under regulation of P1P2 and P4 promoters in different Bt backgrounds

To evaluate how upstream sequences influence Cry toxin expression and affect host performance, we cloned the *cry1Ab* (lepidopteran-specific) and *cry4Ba* (dipteran-specific) genes downstream of either the *cry1Ab* promoter region (P1P2 promoter, 370 bp upstream region) or *cry4Ba* promoter region (P4 promoter, 443 bp upstream region). The resulting constructs were introduced in two different non-toxic, acrystalliferous *B. thuringiensis* strains: Bt subsp. *thuringiensis* (Btt) 407 Cry⁻, whose parental strain is active against lepidopteran larvae [40], and Bt subsp *israelensis* (Bti) 4Q7 Cry⁻, whose parental strain is toxic to dipteran larvae [41]. Growth curve analyses in sporulation HCT medium showed that all transformed strains exhibited comparable growth rates and similar sporulation kinetics (Fig 2A).

Transcription of the *cry* gene was first analyzed by RT-qPCR after 12 and 24 h of growth in sporulation HCT-erythromycin medium (Fig 2B). Both promoters successfully drove expression of *cry1Ab* and *cry4Ba* genes in the two transformed strains. However, overall transcript levels were consistently higher in Btt 407 Cry⁻ strain than in Bti 4Q7 Cry⁻ strain. These differences were not due to the differences in copy number per cell of the plasmids used, since pHT315, used to clone *cry* genes under regulation of P1P2 (P1P2-Cry1Ab and P1P2-Cry4Ba) display 15 copies per cell while pHT611, used to clone *cry* genes under regulation of P4 (P4-Cry1Ab and P4-Cry4Ba) display four copies per cell [42].

For *cry1Ab,* transcript levels after 24 h of growth in HCT medium in the Btt strain were approximately 2.5-times higher than in Bti under same P1P2 regulation. The highest transcription levels of *cry1Ab* gene were observed under regulation of P4 in Btt at 24 h of growth, which were more than seven-fold higher than those observed in Bti under same P4 regulation at 24 h. The lowest *cry1Ab* expression levels were observed in Bti under P4 control. For *cry4Ba,* transcript levels were approximately five-fold higher in Btt than in Bti after 24h of growth when driven by P1P2, whereas under P4 regulation the difference was more modest (approximately 1.5-fold). The lowest *cry4Ba* transcript levels were detected in Bti under P1P2 regulation. Together, these data indicate that both promoter regions are functional in the two Bt backgrounds, but their relative transcriptional activity depends on the specific promoter-gene-strain combination.

Protein accumulation was subsequently analyzed at 8, 12 and 24 h of growth in sporulation HCT medium (Fig 2C). Both Cry1Ab and Cry4Ba proteins were clearly detected by western blot analysis after 24 h of growth, although low levels of Cry4Ba were already detectable at 8 h of growth. Cry1Ab was produced under both P1P2 and P4 regulation, with higher accumulation in the Btt background than in Bti. It is worth to mention that in some cases the SDS-PAGE analysis and western blot analysis did not directly correlate, for example the densitometric analysis (Fiji, ImageJ) of western blots showed that Cry1Ab levels in Bti under P4 regulation were approximately threefold lower than in Btt under the same promoter, consistent with the transcription data. However, SDS-PAGE analysis revealed a smaller difference (approximately twofold) between these two samples, indicating that protein levels estimated by SDS-PAGE analysis and western blot are not directly proportional. This discrepancy likely reflects methodological differences, as SDS-PAGE profiles may include multiple proteins of similar molecular weight, whereas western blots analysis specifically detects proteins recognized by the antibodies used.

For Cry4Ba, both P4 and P1P2 promoters supported protein production in both strains. The highest protein accumulation was observed under P4 regulation in Btt, whereas the lowest levels were found in Bti under P1P2 regulation (nearly two and a half-fold lower in Bti-P1P2 than in Btt-P4 based on western blot analysis).

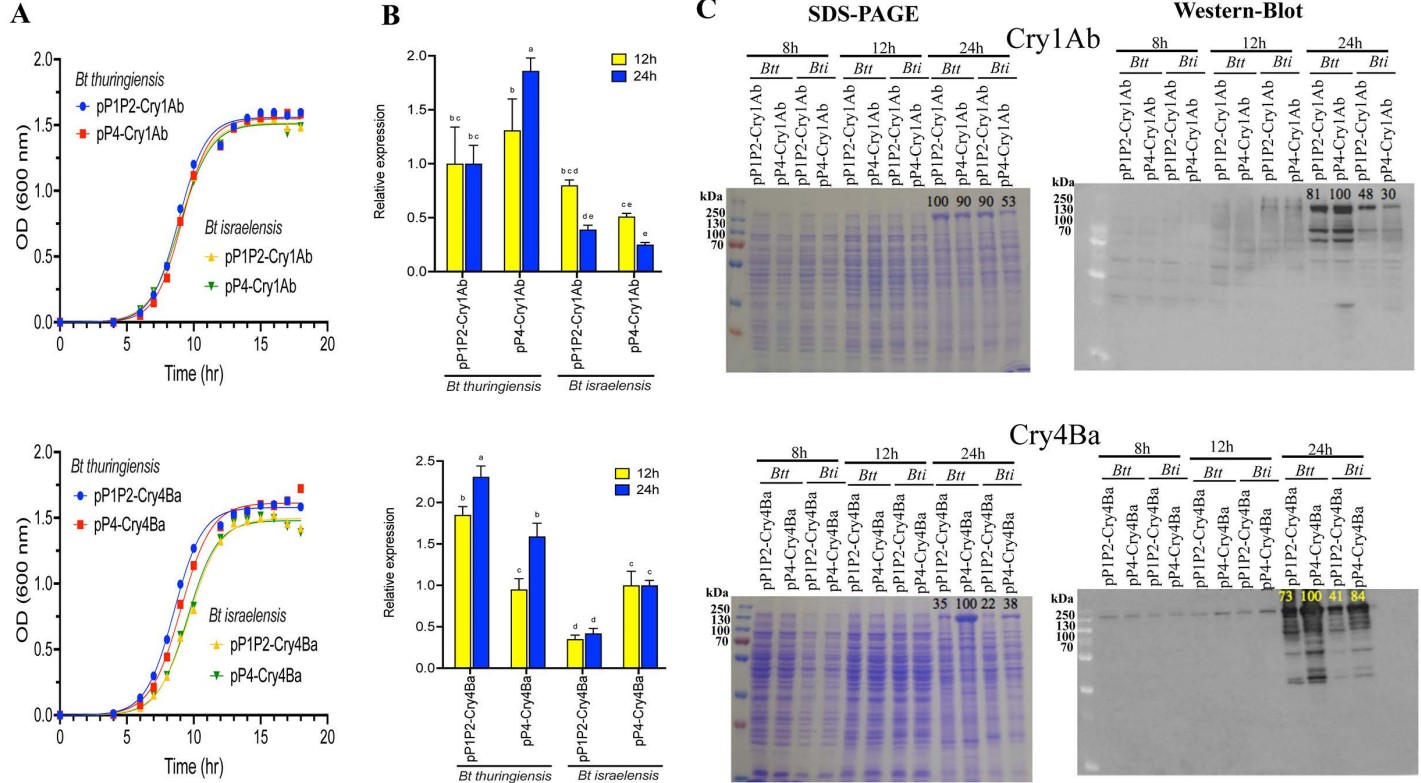

**Fig 2. Expression of Cry1Ab and Cry4Ba under different promoter regions in two *Bacillus thuringiensis* strains.** The *cry1Ab* and *cry4Ba* genes were cloned under the control of P1P2 (370 bp upstream region) or P4 (443 bp upstream region) promoters in *B. thuringiensis* subsp. *thuringiensis* (Btt) 407 Cry⁻, a lepidopteran-specific strain, and *B. thuringiensis* subsp. *israelensis* (Bti) 4Q7 Cry⁻, a dipteran-specific strain. Panel A, growth curve in liquid HCT-erythromycin medium of all strains expressing Cry1Ab (upper panel) or Cry4Ba (lower panel). Panel B, Relative gene expression after 12 and 24 h of growth in this medium determined by RT-qPCR and normalized to the *gatB-YqeY* reference gene (n = 3, performed in triplicate). Data were analyzed using the $2^{\Delta\Delta CT}$ method. Expression of *cry1Ab* under its native P1P2 promoter in Btt and *cry4Ba* under its native P4 promoter in Bti were each set to 1 for normalization. Panel C, Protein production after 8, 12, and 24 h of growth in HCT-erythromycin sporulation medium. A total of 10 µg protein was loaded per lane on SDS-PAGE and analyzed by Coomassie blue staining or by western blot using specific anti-Cry1Ab and anti-Cry4Ba antibodies.

Taken together, these data demonstrated that under controlled laboratory conditions, where both Bt strains exhibit comparable growth patterns and sporulation kinetics, both upstream regions can drive heterologous *cry* genes expression in the distinct Bt genetic backgrounds. The promoter-strain combinations associated with the lowest transcript levels also tended to show the lowest protein accumulation, although the relationship between transcriptional and translational outputs was not strictly proportional across all conditions. The data indicate that P1P2 and P4 are broadly functional across both Bt lineages.

## 2.5. Toxicity of purified spores expressing *cry1Ab* and *cry4Ba* genes under the regulation of P1P2 and P4 promoters in different *Bt* strain backgrounds

To evaluate the functional consequences of upstream regulatory regions, two different *cry* genes were cloned under regulation of either the P1P2 (370 bp) or P4 (443 bp) promoter regions, as explained above and expressed in the non-toxic *B. thuringiensis* subsp. *thuringiensis* 407 Cry⁻ (Btt 407) or *B. thuringiensis* subsp. *israelensis* 4Q7 Cry⁻ (Bti 4Q7) strain backgrounds. Purified spores from the different constructs were used for larval bioassays to assess *in vivo* toxicity. The quality of the spore preparation and absence of Cry1Ab or Cry4Ba proteins prior larval feeding were confirmed by western blot

analyses (S2 Fig). Several studies have reported that Cry proteins can be associated with the surface of *B. thuringiensis* spores [43–45]. Accordingly, we increased the exposure time of the western blot analyses and were unable to detect Cry protein, only very low levels of Cry4Ba protein were found in the spore preparations of Btt strain (S2 Fig).

In lepidopteran bioassays, neonate larvae of *Manduca sexta* and *Spodoptera frugiperda* were fed for 12 h with purified spores and subsequently transferred to clean diet, where they were monitored for several days to record mortality. Larvae exposed to spores from control non-toxic strains Btt Cry⁻ 407 and Bti Cry⁻ 4Q7 showed slight growth delay compared with the water-fed larvae (particularly in *S. frugiperda* bioassays), but no mortality was observed in either lepidopteran species (S3 Fig). Mortality in both *M. sexta* and *S. frugiperda* was highest when larvae were fed with Btt spores expressing *cry1Ab* under P1P2 promoter, whereas the lowest mortality occurred with Bti spores expressing *cry1Ab* under P4. Under P4 expression in Bti, mortality was approximately fourfold lower in *M. sexta* and sixfold lower in *S. frugiperda* compared with the P1P2 driven expression in the Btt background (Fig 3). Since control spores lacking *cry1Ab* gene did not cause mortality, the observed lethality in the experimental groups is consistent with *in vivo* production of Cry1Ab protein.

Conversely, in dipteran assays, *A. aegypti* larvae were most strongly affected by Bti spores expressing *cry4Ba* under the P4 promoter, whereas the weakest effect (more than 25-fold lower value) was observed with Btt spores expressing *cry4Ba* under P1P2 (Fig 3). Interestingly, these *in vivo* toxicity results did not correlate with the *in vitro* expression patterns described above (Fig 2). For example, the highest accumulation of Cry4Ba protein *in vitro* was observed when *cry4Ba* gene was expressed in Btt under P4 regulation after 24 h growth in HCT medium. However, *A. aegypti* larvae exposed to this construction showed relatively low mortality, approximately five-fold lower than that observed with Bti expressing *cry4Ba* under the same promoter. Also, it was shown that Bti expressing *cry4Ba* under P1P2 promoter accumulated the

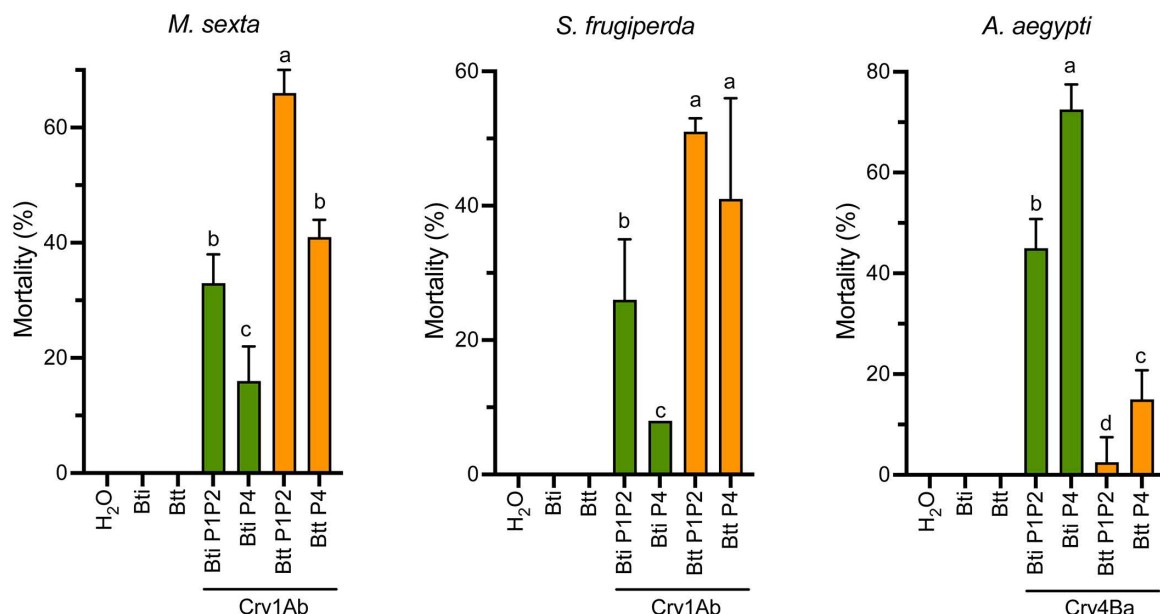

**Fig 3. *In vivo* activity of recombinant Btt and Bti strains against *Manduca sexta, Spodoptera frugiperda* and *Aedes aegypti* larvae.** Lepidopteran bioassays were performed with first-instar *M. sexta* or *S. frugiperda* larvae on artificial diet treated with 100,000 spores per well from Btt or Bti cells transformed with either pP1P2-*cry1Ab* (370 bp upstream region) or pP4-*cry1Ab* (443 bp upstream region). After 12 h feeding, larvae were transferred to fresh clean diet and mortality was recorded after five days. For *A. aegypti* bioassays, second-instar larvae were exposed to 100,000 spores from transformed Btt or Bti strains (pP1P2-*cry4Ba* or pP4-*cry4Ba*) in 10 mL water. All bioassays were done in triplicate. Negative controls (H₂O or purified spores from untransformed strains) showed no mortality. Statistical significance was determined by one-way ANOVA; different letters indicate significant differences at $p < 0.05$.

lowest levels of Cry4Ba protein in HCT cultures, yet this strain caused substantial larval mortality *in vivo* 2.5- and 5-fold higher than that caused by Btt spores expressing *cry4Ba* under P4 or P1P2, respectively. Together, these discrepancies indicated that *in vitro* expression levels of Cry proteins do not necessarily predict *in vivo* toxicity. Rather, the bioassays revealed that highest mortality was induced by Btt P1P2 construct in lepidopteran larvae and by the Bti P4 construct in dipteran larvae. Multiple biological factors are likely to contribute to these outcomes, including differences in spore germination, bacterial growth, and toxin production within the insect midgut. Also, expression of additional virulence factors cannot be discarded. It is therefore reasonable to propose that different *Bt* subspecies have been adapted to germinate and proliferate in their respective host insects, and that the observed differences in toxicity reflect complex host–strain interactions and promoter-dependent Cry protein expression is just an additional factor.

## 3. Discussion

Cry proteins from *Bacillus thuringiensis* (Bt) constitute one of the largest, and extensively studied families of bacterial toxins. While they clearly evolved from a common ancestor, their upstream regulatory regions show striking heterogeneity. Most *cry* genes are expressed during the sporulation phase, under the control of $\sigma^E$ and $\sigma^K$ factors [6–11], yet comparative analyses have revealed substantial variability in their upstream sequences [23,24,30–38,46–49]. This raises the question of why *Bt* maintains such diverse regulatory strategies for genes encoding structurally similar toxins and biological functions with overlapping host specificities?.

Previous studies demonstrated that *cry* genes are often plasmid-borne and strongly influenced by horizontal gene transfer and recombination, frequently mediated by insertion sequences (IS) (e.g., IS231, IS232, IS240, ISBt1, and ISBt2), and transposons (Tn) (e.g., Tn4430 or Tn5401) [47]. Indeed, genomic analysis identified pathogenicity islands (PAIs) that cluster toxins with similar host ranges [46,48], reflecting co-evolution of genomic architecture and ecological adaptation. For example, the BtPAI-I to BtPAI-III, prevalent in lepidopteran-active strains such as *B. thuringiensis* subsp. *aizawai, kurstaki, galleriae, tolworthi*, *colmeri, amagiensis*, and *azorensis*, contain *cry*1, *cry*2 and *vip*3 genes, all of which encode toxins active against lepidopteran insects [46,48]. Conversely, the BtPAI-IV and BtPAI-V, predominant in *B. thuringiensis* subsp. *israelensis*, encode *cry4, cry10, cry11*, and *cyt* genes with dipteran specificity [46]. These findings support the idea that gene content and regulatory architecture co-evolved to optimize host specialization in distinct Bt lineages.

Within this framework, our MEME-based analysis provides novel insight into the evolutionary diversification of *cry* upstream regions. Due to the low sequence similarity classical phylogenetic reconstruction cannot be done. For this reason, we used MEME analysis to identify conserved, alignment-independent motifs, revealing 24 motif blocks that define distinct regulatory architectures in the upstream regions of *cry* family. The organization of these motifs often correlates with the insect host targeted by each toxin, suggesting that regulatory elements have co-evolved with host specificity. However, different architectures were selected in each insect order, indicating a high flexibility of Bt evolution and we highlighted several cases of *cry* genes encoding highly similar proteins that showed divergent evolution in their upstream regions, indicating regulatory decoupling and supporting evolutionary flexibility of regulatory regions.

Genomic studies have shown that the *cry1A* genes, not only shared conserved upstream motifs but are also consistently preceded by two additional genes encoding an amidase and a cation antiporter [49]. This conserved genomic context suggests a shared evolutionary history and possibly an adaptive role in insect-specific environmental conditions, such as an alkaline midgut lumen. Our analyses confirm that the upstream sequences of *cry1* genes, except *cry1I*, are strongly conserved. However, even within *cry1* group, expression strength can vary markedly. For instance, *lacZ* gene fusion assays revealed that upstream regions of *cry1Da*, *cry1Ae*, *cry1Bb*, *cry1Fb*, and *cry1Ja*, all containing the P1P2 promoter, differ substantially in β-galactosidase activity, implying that $\sigma^E$, and $\sigma^K$ regulation is further modulated by additional cis- or trans- acting factors [50].

Interestingly, our analysis also showed that some *cry* genes appear to have diverged more radically. The *cry1I* group is one such case: unlike most Cry1, the Cry1I proteins are secreted rather than crystallized in parasporal inclusions [30,51],

these genes show no similarities in their regulatory architecture of their upstream regions with other *cry1* genes. As mentioned above, the *cry1I* gene was previously proposed to be transcribed by the sporulation sigma factor σ^E [30]. However, re-examination of primer extension data reported by Tounsi and Jaoua [30] reveals multiple upstream transcriptional signals, suggesting that additional transcription start sites may be located further upstream than those originally proposed. Moreover, the spacing between the predicted −35 and −10 elements of the putative σ^E-dependent promoter is inconsistent with the canonical σ^E-dependent promoters, arguing against direct σ^E control. Consistent with this former argument, a previous publication showed that Cry1I is produced earlier during stationary phase than Cry1A-type toxins [51], and recent evidence indicates that *cry1I* is regulated by VipR, the same transcriptional activator that controls *vip3A* expression [39]. In agreement with this model, we identified the predicted VipR-binding motif reported by Chen et al 2022 [39] not only upstream of *cry1Ia*, but also in the upstream regions of *cry9Ca-orf1*, *cry2Ab*, and *cry9Aa* genes (S4 Table), suggesting that these genes may be subject to an additional regulatory control that remains to be characterized. Together, these observations indicate that *cry1I* belongs to a distinct regulatory class and that its transcriptional regulation warrants further refinement.

Similarly, *cry2* genes form a subgroup with specific divergence: *cry2Ac* and *cry9Ca* share nearly identical upstream sequences distinct from *cry2Ab* and *cry2Ad*, while *cry2Aa* shares partial similarity with the upstream regions of *cry2Ac* and *cry9Ca* genes. Notably, Cry2Aa and Cry2Ab differ not only in their upstream architecture but also in their host specificity, since Cry2Aa is active against both dipteran and lepidopteran insects, while it is well stablished that Cry2Ab is restricted to lepidopteran [23] and is not active against *Aedes aegypti,* although it has been reported to have some dipteran activity [52]. These examples illustrated that even closely related toxins have evolved distinct regulatory mechanisms, likely contributing to their ecological versatility.

Our findings thus expand the view that *cry* gene expression is not uniformly controlled by σ^E and σ^K factors. Upstream regulatory elements are highly variable, even among toxins targeting the same insect orders. It is possible that certain insect orders exhibit different midgut conditions among different species, for instance coleopteran weevil's vs beetles or dipteran mosquitoes vs flies, reflecting diverse regulatory regions for Cry proteins targeting different hosts. Thus, further research is also needed to clarify the functional contributions of these regulatory elements to Cry expression efficiency across Bt lineages, and host environments.

Post-transcriptional regulation also emerges as an important layer of control. The *cry3Aa* gene is a classic case, whose transcription initiates far upstream (position -558), generating a long primary transcript that is processed into a shorter, stable mRNA form protected by a Shine–Dalgarno-like (STAB-SD) sequence [31,32]. Our FIMO analysis indicates that *cry3Ca* shares this feature (position -125) and MEME-block motives analysis revealed that the upstream regions of *cry3Ba* and *cry3Bb* resemble each other but diverge from *cry3Aa*. The presence of additional Shine-Dalgarno (SD) motifs in the upstream region of multiple *cry* genes (*cry1Ba, cry1Ia, cry1Id, cry1Ka, cry2Aa, cry2Ab, cry2Ac, cry2Ad, cry3Aa, cry3Bb, cry4Ba, cry8Hb, cry8Kb, cry8La, cry9Aa, cry9Ca, cry10Aa, cry19Aa, cry26Aa*, *cry29Aa, cry32A, cry41Aa1* and *cry61Aa*) suggest possible roles in transcript stability or translations efficiency, although these functions have not yet been systematically explored.

In addition to *cry3A* gene, other *cry* genes also exhibit distant promoter positions relative to the start codon. For example, the σ^E dependent *cry4A* promoter is located -364 bp, while the σ^K dependent promoter is found in position -654 bp [15]. However, unlike *cry3Aa*, these genes lack a putative STAB-SD region.

Operon organization adds yet another regulatory complexity. Several *cry* genes occur in operons alongside accessory ORFs that influence toxin crystallization or stability [24,53]. In the *cry2Aa* and *cry2Ac* operons, proteins with repetitive motives encoded by adjacent ORFs may be acting as scaffolds for crystal formation [24]. Similarly, the *cry4B, cry11Aa* and *cry10Aa* operons encode additional small proteins such as p19 and p20, which act as chaperone-like factors [53]. These examples underscore that *cry* gene expression cannot be fully understood in terms of promoter activity alone, since operon context and accessory proteins that have co-evolved with the toxins may also play key roles in optimizing toxin expression and function.

Taken together, our analyses converge on the conclusion that Cry family of proteins exhibit remarkable diversity in their regulatory-elements architecture. The plasmid-borne distribution of *cry* genes within PAIs, subgroup-specific upstream motifs, operons-associated accessory proteins, and stabilizing elements such as STAB-SD, collectively illustrate the multiplicity of strategies by which Bt has optimized Cry toxin expression. We propose that this regulatory variability is not accidental, but reflects selective pressures to maximize toxin production under the physiological conditions specific to different insect hosts. This evolutionary fine-tuning provides a compelling explanation for the ecological success and host specialization of Bt.

Our *in vitro* and *in vivo* experiments further support these ideas as a proof of concept. Together, they demonstrate that the efficacy of Bt Cry toxins in natural settings is determined not only by the intrinsic properties of the toxins themselves, but also by the specific combination of upstream regulatory sequences and bacterial strain background, which jointly influenced multiple factors that finally affect *in vivo* toxin expression levels, and insecticidal performance.

*In vitro* assays confirmed that both promoters were functional in Btt and Bti under optimal sporulation conditions after culture in HCT medium. However, under these growth conditions, transcript levels and protein accumulation were generally higher in Btt than in Bti, Specifically, Cry4Ba production was lowest in Bti when expressed under P1P2, whereas Cry1Ab production was lowest in Bti under P4 promoter. Further studies involving a broader range of Bt strains and upstream regulatory regions will be necessary to stablish general patterns of Cry protein expression under *in vitro* culture conditions.

*In vivo* bioassays revealed that, in lepidopteran larvae, the highest mortality occurred when *cry1Ab* was expressed under P1P2 promoter in Btt background. In contrast, *A. aegypti* larvae were most strongly affected when *cry4Ba* was expressed under P4 promoter in Bti background, followed by P1P2-driven expression in Bti. Interestingly, these *in vivo* effects did not correlate with *in vitro* protein accumulation, suggesting that additional biological factors within the larval midgut contribute to toxicity. Such factors likely include spore germination, bacterial growth, and toxin production within the insect midgut environment.

Previous studies have demonstrated that Bt spores can germinate within the insect gut. However, these analyses were largely performed using larvae that ingested spore–crystal suspensions [54–56]. In contrast, relatively few studies have specifically examined the effects of purified spores from toxic strains on midgut tissues in the absence of crystal inclusions. Wilson and Benoit [57] reported that Bt spores are able to germinate in *M. sexta* midgut fluid, indicating that gut conditions alone can support spore germination. In addition, Sutter and Raun [58] showed that purified spores from a native toxic strain can induce vacuolation in midgut tissues.

Our data are consistent with the notion that different *Bt* subspecies may be evolutionarily adapted to germinate and proliferate more efficiently in their respective host insects. The fact that the acrystalliferous Bt strains alone did not cause mortality in either lepidopteran or dipteran larvae further highlights the essential role of Cry toxin expression for Bt pathogenicity.

It is important to note, that in addition to the loss of *cry* toxin genes in the non-toxic Btt 407 and Bti 4Q7 strains, these non-toxic acrystalliferous strains have also lost other plasmid-encoded genes associated with virulence. Consequently, the derivative strains (Btt 407 and Bti 4Q7 used as negative controls and also as recipients for the different plasmids expressing Cry1Ab or Cry4Ba toxins) may differ from their original parental toxic strains in aspects beyond Cry toxin production and may not fully represent all virulence-related traits of the wild-type isolates.

Although additional experiments with alternative promoter regions and other Bt strains are desirable, our results may indicate that Cry efficacy depends not solely on promoter strength, but on the specific combination of promoter and strain background: P1P2-driven expression in Btt maximized Cry1Ab toxicity against susceptible lepidopterans, whereas P4-driven expression in Bti optimized Cry4Ba activity against mosquitoes. Upstream regulatory regions are therefore not universally interchangeable, but have been selectively optimized to function under the ecological and physiological conditions of their native hosts. This supports the hypothesis that the diversity among *cry* upstream sequences represents an evolutionary adaptation to maximize toxin efficacy and bacterial fitness across distinct insect environments.

Overall, our findings highlight how promoter-strain interactions and host specialization have co-evolved to fine-tune Cry toxin expression, providing a framework for rational optimization of Bt strains for targeted insect control and for understanding the evolutionary pressures shaping toxin regulation.

## 4. Methods

### 4.1. MEME and FIMO analyses of *cry* genes upstream sequences

The upstream sequences of *cry* genes were obtained from the BPPRC database (S1 Table). For comparative analysis, sequences of comparable length of 170–250 bp upstream of the ATG start codon were selected, except for *cry4Aa* (404 bp) and *cry4Ba* (316 bp), which were extended to include their mapped promoter regions (S2 Table). Conserved motif blocks were identified using the "Multiple Expectation Maximization for Motif Elicitation" (MEME) algorithm (https://meme-suite.org/meme/doc/cite.html) [20] applied either to the complete upstream sequences of *cry* genes listed in S1 Table or to the subset of upstream regions with comparable lengths (170–250 bp) shown in S2 Table. The individual occurrences of conserved motifs were subsequently verified with the "Find Individual Motif Occurrences" (FIMO) tool (https://meme-suite.org/meme/tools/fimo) [29] using the complete upstream sequences reported in S1 Table. FIMO analyses included searches for promoter regions previously described for specific *cry* genes, as well as for Shine–Dalgarno (SD) sequences and other motifs mentioned in the Introduction.

For both MEME and FIMO analyses, the reported *p*-values represent the probability that a random sequence of the same composition and length would show a motif match with an equal or higher log-likelihood ratio at a given position, under the null model of random sequence generation. In this study, $p < 0.001$ was considered highly significant.

### 4.2. Cloning of *cry1Ab* and *cry4Ba* genes under different promoter regions

The *cry1Ab* gene was amplified by PCR using pHT315-*cry1Ab* as template [40,59] and primers containing *NcoI* and *XmaI* restriction sites at the 5' and 3' ends, respectively (S5 Table). In plasmid pHT315-*cry1AbMod* [60], which carries a 370 bp upstream region of *cry1Ab* (including the P1P2 promoters), the *cry1AbMod* gene was replaced with the PCR-amplified *cry1Ab* gene, generating plasmid pP1P2-*cry1Ab*.

To construct *cry1Ab* under the *cry4Ba* promoter, the plasmid pHT611-*cry4Ba* [42] containing a 443 bp upstream region was modified by site-directed mutagenesis using mutagenic primers (S5 Table). These modifications included the introduction of a *NcoI* restriction site at the start codon (position 1) and a *XmaI* site 285 bp downstream of the stop codon. Additionally, two internal *NcoI* sites located in the terminator region of *cry4Ba* (at positions 30 bp and 48 bp downstream of the stop codon) were removed. The *cry4Ba* coding sequence was then replaced with the *cry1Ab* gene amplified with *NcoI* and *XmaI* restriction sites, yielding plasmid pP4-*cry1Ab*. The unmodified pHT611-*cry4Ba* plasmid (with its 443 bp upstream region) was designated pP4-*cry4Ba* in this study.

For expression of *cry4Ba* under the P1P2 promoter, the *cry4Ba* gene was PCR-amplified from the modified pHT611-*cry4Ba* plasmid using primers containing *NcoI* and *XmaI* sites (S5 Table). The amplified fragment replaced *cry1AbMod* in pHT315-*cry1AbMod* producing plasmid pP1P2-*cry4Ba.*

The resulting plasmids (pP1P2-*cry1Ab*, pP1P2-*cry4Ba*, pP4-*cry1Ab,* and pP4-*cry4Ba*) were checked by DNA sequencing and transformed into two acrystalliferous *Bt* strains: *Bt* subsp. *thuringiensis* 407 serotype H1 [40] and *Bt* subsp. *israelensis* (Bti) strain 4Q7 [41] obtained from the *Bacillus* Stock Center at Ohio State University (Columbus, OH).

### 4.3. Expression and immunodetection of Cry1Ab and Cry4Ba toxins

Recombinant Bti 4Q7 and Btt 407 strains transformed with pP1P2-*cry1Ab*, pP1P2-c*ry4Ba*, pP4-*cry1Ab*, and pP4-*cry4Ba* plasmids were grown over night (o.n.) in LB Petri dishes supplemented with 10 µg/mL erythromycin to have single colonies. A single colony was then selected and grown again o.n. in LB Petri dishes supplemented with 10 µg/mL erythromycin

to have again single colonies. One colony was selected and cultured in 5 ml of liquid LB supplemented with 10 µg/mL erythromycin for 12 h. After this time this preculture was diluted 1:20,000 in 50 ml of HCT sporulation medium [61] supplemented with 1% glucose and 10 µg/mL erythromycin. Samples were collected after 8, 12, and 24 h of incubation and washed three times with 300 mM NaCl, 10 mM EDTA (pH 8), followed by three washes with 1 mM phenylmethylsulfonyl fluoride (PMSF) to inhibit proteases. Protein concentrations were determined by the Bradford assay (Bio-Rad) using bovine serum albumin (BSA) as a standard. Protein integrity and expression levels were analyzed by SDS-PAGE (10 µg protein were loaded in each lane) with Coomassie blue staining and analyzed by using imageJ program [62].

For western blot analysis, proteins were separated by SDS-PAGE and transferred to a PVDF membrane at 350 mA for 45 min. Membranes were blocked for 1 h at room temperature with 5% nonfat dry milk in PBS, then incubated for 1 h with anti-Cry1Ab or anti-Cry4Ba primary antibodies (1:25,000 dilution). After three washes with PBS–T (PBS with 0.1% Tween 20), membranes were incubated for 1 h with HRP-conjugated mouse anti-rabbit secondary antibody (1:20,000 dilution). The membranes were washed three times with PBS-T and detection was performed using Luminol SC-2048 (Santa Cruz Biotechnology) following the manufacturer's instructions, and signals were visualized with an Amersham Imager 600 (GE Healthcare Life Sciences, Little Chalfont, UK). All experiments were done in triplicate.

### 4.4. Analysis of *cry1Ab* and *cry4Ba* gene expression by RT-qPCR

Transformed *Bt* strains were cultured in HCT sporulation medium as described above, then 3 and 1.5 mL samples were collected after 12 and 24 h of growth, respectively, for total RNA extraction using the Quick-RNA MiniPrep kit (Zymo Research). RNA concentration and purity were assessed with a Nanodrop 2000c spectrophotometer (Thermo Scientific).

First-strand cDNA synthesis was performed with SuperScript III Reverse Transcriptase (Invitrogen). RT-qPCR assays were carried out in triplicated with 500 ng of cDNA using a Maxima SYBR Green/ROX qPCR Master Mix (Thermo Scientific) and gene-specific primers (0.15 µM each; S5 Table) in a 48-well plate format. The *gatB-YqeY* gene was used as a reference gene [63]. Amplification was performed in an Eco real-time PCR system (Illumina) and data analysis (melting curve evaluation and relative quantification using the $2^{-\Delta\Delta CT}$ method), was done with EcoStudy software (Illumina).

For normalization, the expression of *cry1Ab* under its native P1P2 promoter in Btt, and expression of *cry4Ba* under its native P4 promoter in Bti were both set to 1. All assays were performed in triplicate.

### 4.5. Purification of spores

The transformed *Bt* strains, along with acrystalliferous Btt 407 and Bti 4Q7 strains used as negative controls, were cultured each in three Petri dishes containing HCT sporulation solid medium (supplemented with 10 µg/mL erythromycin for the transformant strains) for 72 h or until complete sporulation was achieved, as confirmed by microscopic observation. It has been reported that Bt spores exhibit moderate to highly hydrophobicity nature and that their hydrophobicity is significantly greater than that of their corresponding vegetative cells or parasporal crystals [64,65]. We took advantage of this property to purify the spores. The cultures in HCT-erythromycin were recovered using a bacteriological loop in sterile conditions, vigorously vortexed 3–5 min in 15 ml of sterile 3 M NaCl and 0.01M EDTA solution, and washed three times by centrifugation at 10,000 rpm for 10 min in plastic sterile tubes (Nalgene PPCO centrifuge tube with screw cap). If foam was formed, it adheres to the plastic tube. We recover only the supernatant, by decanting very slowly to avoid incorporating foam into the sample. Then, before adding the washing solution again, we rinse the tube with a small amount of sterile water to remove the foam and discard. The supernatant from each wash, containing the floating spores, was collected. The combined supernatants (~45 mL) were ultracentrifuged at 23,000 rpm for 30 min at 15°C to recover the spores. The resulting pellets were washed three times with sterile Milli-Q water and finally resuspended in sterile Milli-Q water.

Spore concentration was determined by plating serial dilutions ($10^{-1}$ to $10^{-11}$) on LB agar supplemented with 10 µg/mL erythromycin for transformant strains or without antibiotic for control strains, and counting colony-forming units (CFU) after o.n. incubation at 30°C. Sample quality and the absence of crystal proteins were verified by microscopic examination

and by loading 20,000 spores per lane on SDS-PAGE (10% acrylamide gel), followed by western blot analysis using anti-Cry1Ab or anti-Cry4Ba antibodies (S2 Fig). Purified spores were stored at 4°C until use.

### 4.6. Insect bioassays with purified spore samples

Bioassays were performed using *M. sexta* or *S. frugiperda* larvae reared on artificial diet [66]. The diet surface in 24-well plates was treated with 100,000 spores/well from Btt or Bti strains transformed with pP1P2-*cry1Ab* or pP4-*cry1Ab* plasmids. One first instar larvae was placed per well. For each toxin concentration treatment, one 24-well plate was tested in triplicate (72 larvae per treatment), and five toxin concentrations were used per sample. Plates were incubated at 28°C with 65% ± 5% relative humidity, under a 16 h light/8 h dark cycle photoperiod. After 12 h of feeding, the larvae were transferred to fresh diet and maintained under the same conditions. No spore germination was detected, as evidenced by the absence of bacterial growth on the diet surface since no Bt colony formation was registered. The diet contaminated with spores remained unchanged even after several days of incubation under the same conditions used for the bioassays (up to 10 days). Larval mortality was recorded five days after treatment. Negative controls included diet treated with spores from acrystalliferous Btt 407 or Bti 4Q7 strains or with sterile water.

All experiments were conducted in triplicate. Statistical significance was determined by one way ANOVA, using Graph-Pad Prism software, where different letters indicate significant differences with $p$ value < 0.05; no significant differences are indicated with same letter or as "ns".

For dipteran bioassays, groups of ten second instar *A. aegypti* larvae were placed in 10 mL of water containing 100,000 spores from Btt or Bti strains transformed with pP1P2-*cry4Ba* or pP4-*cry4Ba* plasmids. Controls included spores from acrystalliferous Btt 407 and Bti 4Q7 strains. No mortality was observed in the control treatments and spores did not germinate in water. After 48 h of incubation, larval mortality was recorded.

## Supporting information

**S1 Table. Upstream sequence reported in BPPRC for 82 *cry* genes (one representative gene selected from each *cry* gene subgroup) (FASTA format).**
(DOCX)

**S2 Table. Selected upstream sequences for *cry* genes with comparable length (170–250 bp) (FASTA format).**
(DOCX)

**S3 Table. Sequence alignment of block motifs identified by MEME analysis.** The reported $p$-values represent the probability that a random sequence of the same composition and length would show a motif match with an equal or higher log-likelihood ratio at a given position, under the null model of random sequence generation. Values of $p < 0.001$ were considered highly significant. Gray shading in the sequences indicates regions with high identity to promoter elements (-10 and -35 regions) or Shine-Dalgarno (SD) elements as indicated in each block alignment.
(DOCX)

**S4 Table. Results of FIMO searches for individual motif occurrences in *cry* gene upstream regions.** FIMO ("Find Individual Motif Occurrences") analysis was used to locate individual matches of FIMO-identified motifs within the complete upstream sequences of *cry* genes. The $p$-value for each motif occurrence represents the probability that a random sequence of equivalent composition would achieve a motif score (log-likelihood ratio) equal to or greater than that observed at a given position. Values of $p < 0.001$ were considered highly significant.
(DOCX)

**S5 Table. Oligonucleotide sequences used in this study.**
(DOCX)

**S1 Fig. MEME-identified motif-blocks in the upstream sequences from 82 *cry* genes.**
(TIF)

**S2 Fig. Western blot analysis of purified spores.** Panels A and C, Western blot of Cry1Ab producing strains analyzed with anti-Cry1Ab antibody. Panels B and D, Western blot of Cry4Ba producing strains analyzed with anti-Cry4Ba antibody. Panels C and D, images after increasing the exposure time of the western blot. The acrystalliferous Btt and Bti strains were included as negative controls.
(TIF)

**S3 Fig. Bioassays of *Spodoptera frugiperda* and *Manduca sexta* larvae treated with H$_2$O, or purified spores from Btt407 and Bti4Q7 control strains.** Panel A, imagen of representative larvae. Panel B, larval weight in gr after five days treatment with Btt or Bti spores. H$_2$O was included as negative control.
(TIF)

## Acknowledgments

We thank Javier Luévano Borroel from CINVESTAV-Irapuato for providing *Manduca sexta* larvae.

## Author contributions

**Conceptualization:** Mario Soberón, Alejandra Bravo.

**Formal analysis:** Oscar Infante, Mario Soberón, Alejandra Bravo.

**Funding acquisition:** Alejandra Bravo.

**Investigation:** Isabel Gómez, Blanca I. García-Gómez, Nathaly A. do Nascimento, Oscar Infante, Pablo Emiliano Cantón, Sabino Pacheco, Angel E. Peláez-Aguilar, Jorge Sanchez, Alejandra Bravo.

**Writing – original draft:** Alejandra Bravo.

**Writing – review & editing:** Mario Soberón.

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
