## [Decision Letter · Decision Letter 0]

14 Jan 2026

Regulatory diversity in Bacillus thuringiensis cry genes reveals flexible evolutionary strategies for in vivo toxin expression

PLOS Pathogens

Dear Dr. Bravo,

Thank you for submitting your manuscript to PLOS Pathogens. After careful consideration, we feel that it has merit but does not fully meet PLOS Pathogens's publication criteria as it currently stands. Therefore, we invite you to submit a revised version of the manuscript that addresses the points raised during the review process.

We look forward to receiving your revised manuscript.

Kind regards,

Jens Rolff

Academic Editor

PLOS Pathogens

Debra Bessen

Section Editor

Editor-in-Chief

PLOS Pathogens

PLOS Pathogens

orcid.org/0000-0002-7699-2064

**Additional Editor Comments:**

We have now three reviews. All reviewers agree that this is an interesting and ambitious manuscript. Reviewer 2 assessed the previous version and I agree that many of the original critical points have been addressed. Reviewer 1 highlights some important points. For any revision it would be very important to adequately address the point, especially the first and the last point.

**Journal Requirements:**

1) We noticed that you used the phrase 'data not shown' in the manuscript. We do not allow these references, as the PLOS data access policy requires that all data be either published with the manuscript or made available in a publicly accessible database. Please amend the supplementary material to include the referenced data or remove the references.

- TM on page: 14.

3) Some material included in your submission may be copyrighted. According to PLOSu2019s copyright policy, authors who use figures or other material (e.g., graphics, clipart, maps) from another author or copyright holder must demonstrate or obtain permission to publish this material under the Creative Commons Attribution 4.0 International (CC BY 4.0) License used by PLOS journals. Please closely review the details of PLOSu2019s copyright requirements here: PLOS Licenses and Copyright. If you need to request permissions from a copyright holder, you may use PLOS's Copyright Content Permission form.

Potential Copyright Issues:

i) Please confirm (a) that you are the photographer of S3, or (b) provide written permission from the photographer to publish the photo(s) under our CC BY 4.0 license.

4) Kindly revise your competing statement in the online submission form to align with the journal's style guidelines: 'The authors declare that there are no competing interests.'

**Reviewers' Comments:**

Reviewer's Responses to Questions

**Part I - Summary**

Reviewer #1: This study investigates the divergence of upstream regulatory regions of Bacillus thuringiensis (Bt) cry genes to identify conserved and non-conserved regulatory elements—such as promoters and ribosome-binding sites—and to determine whether the similarities detected in promoter architecture correlate with host specificity. The authors combine two parallel approaches: a genomic analysis of regions upstream of cry genes and a genetic strategy involving reciprocal exchanges of promoters and coding sequences between two cry genes. This latter approach aims to evaluate gene expression and insecticidal activity under the control of two different promoters and in two Bt strains originally exhibiting distinct host spectra (Lepidoptera versus Diptera).

Overall, the results suggest that cry gene expression and the associated insecticidal activity depend on both promoter origin and strain genetic background, consistent with a fairly logical evolutionary relationship between regulation and ecological function. This is an extremely ambitious study based on substantial work. However, various parts of this work contain highly questionable points, imprecisions and an error due to the unreliability of a publication on which the genomic analysis is based.

Reviewer #2: This revised manuscript has certainly attempted to water down the claims made in the original version and many of my technical queries have been addressed. I found the analysis of the upstream regions interesting and was pleased to see that attempts were made to test resulting hypotheses. Unfortunately, I still feel that the manuscript tries too hard to fit their data to their primary hypothesis (that upstream elements have evolved in a target-specific way). Two examples of this are:

1) Lines 369-370 state that there was a consistent relationship between transcriptional and translational outputs. This seems to have been extrapolated from the previous (essentially true) statement that the lowest transcript levels showed the lowest expression, but does not hold true for all the data. For example: qPCR shows that expression of Cry1Ab from P4 in Btt is around 4x that of Cry1Ab expressed from P1P2 in Bti. The SDS-PAGE gel shows that the amount of protein produced by the two clones is the same. Strangely the western blot indicates that one clone produced twice as much protein as the other. While there are obvious reasons why the transcript and protein levels may not correlate, I was surprised that the authors did not comment on the disparity between the SDS PAGE and western blot data.

2) Lines 405-406 state that the data indicates that the midgut environment modulated the Cry protein’s performance. I am not sure whether the authors mean that the environment affected the expression or the activity of the expressed protein. Assuming the former, then it is still a big leap, many factors are involved in the process leading up to toxicity including the ability of the spore to germinate in the insect. It is not unreasonable to think that the different subspecies of Bt may have evolved to germinate (or grow) better in their host insect and so the difference would have no relationship to toxin expression. The data in Fig3 show a much better correlation between host strain and toxicity than between promoter region and toxicity. If the authors are indeed suggesting that the protein (once produced) behaves differently depending on how it was expressed then this needs a lot more discussion.

Aside from the above two points I have the following additional comments.

3) Lines 341-344 are confusing. Although it is stated that 407 and 4Q7 are acrystalliferous the text implies that they have activity against insects. The data in Fig 3 suggests that they don’t. The authors presumably mean that the strains were derived from those with those activities. This should be made clear. It is important as the plasmids in the wild-type strains do encode many genes related to virulence and so derivative strains may not fully represent the parents. This reviewer found the use of Btt confusing since this has traditionally been used to describe B. thuringiensis subsp. tenebrionis. In the context of the manuscript however it is probably OK although I note that in Fig 2A Btt is written as B. thuringiensis whereas it should read Bt thuringiensis.

4) Lines 195-197 highlight the fact that the upstream regions of CryAa/c differ from that of Cry2Ab/d. That is hardly surprising since the region was taken immediately upstream of Cry2Ab/d but much further upstream of Cry2Aa/c (because of the orf1 and orf2 genes). This is discussed later but should be acknowledged here.

5) It is good to see that the authors show that no toxin was present in the spores delivered to the insects. They also state (Lines 653-654) that no spore germination was observed on the diet. No data is shown, which is OK, but we should be told how this was assessed.

6) In the response to reviewers comments it is stated that the larvae were force fed. This would traditionally mean holding up liquid to the larva’s mouth forcing it to imbibe. There is no indication that the larvae were force-fed here.

7) In many places sigma factors have been represented by an “s” rather than the proper Greek character.

Reviewer #3: Interesting work about Bt promoters of Cry genes and its relation with insecticidal activity and their ecological papel.

It could be important for promoter selection in industrial fermentation process to obtain higher yield protein production.

This work is the first systematic analysis of regulatory regions in a bacterial with insect toxin genes

**Part II – Major Issues: Key Experiments Required for Acceptance**

Reviewer #1: 1. The conclusion that the promoter region of cry1I is markedly distinct from those of other cry genes is based on reference #28 (Tounsi and Jaoua, 2002). If the conclusion is true, the explanation provided by the authors is not correct. Inspection of the primer extension data in that study (Ref #28, Fig. 2B) reveals multiple upstream signals, suggesting that the transcription start site is located further upstream than proposed. In addition, the spacing between the predicted −35 and −10 boxes is inconsistent with recognition by the sporulation sigma factor SigE. Previous work has shown that Cry1I is produced early in stationary phase, earlier than Cry1A-type proteins (Ref #44, Kostichka et al., 1996, J. Bacteriol.), and more recent evidence suggests that cry1I expression is regulated by the same factor as vip3A (Chen et al., 2022, Microbiol. Spectr.). Together, these data indicate that the regulatory classification of cry1I warrants further refinement.

2. In Table S4, motif detection using FIMO identifies similarities among putative −35 and −10 boxes; however, internal sequence conservation and genomic context are not consistently taken into account. The presence of short motifs alone (e.g., GGAGG) is insufficient to confidently assign functional regulatory elements such as Shine–Dalgarno sequences. Incorporating positional constraints and conservation of surrounding sequences would strengthen these analyses. For example, Spo0A-like motifs located more than 1 kb upstream of a start codon are unlikely to exert a direct effect on transcription; similarly, Stab-SD sequences positioned at a correct distance (7–11 bp) upstream of potential start codons are more likely to function as RBS.

3. In sections 2.4, 4.2, and 4.4, expression of cry1Ab and cry4Ba is assessed using high-copy-number plasmids, at least for pHT315, and likely also for pHT618, although no information can be found for this plasmid. Since cry genes are naturally carried by low-copy-number plasmids, this experimental design precludes meaningful inference about native expression levels and undermines the interpretation of the results. Furthermore, RT-qPCR measurements performed at 24 h provide only a snapshot of mRNA abundance and do not account for transcriptional dynamics. Protein accumulation data suggest that transcription occurs primarily between 12 h and 24 h. Expression kinetics during this interval should therefore be determined to accurately assess transcriptional efficiency. Importantly, this analysis should be integrated with sporulation kinetics, which may differ between the two Bt strains.

4. The bioassays described in section 2.5 suggest that Bt spores can germinate and that bacteria can proliferate within insect larvae in the apparent absence of Cry toxins. This interpretation contradicts long-standing and widely accepted results demonstrating that toxemia caused by Cry toxins is required for spore germination and bacterial proliferation (Heimpel and Angus, 1959; Du and Nickerson, 1996). To support the conclusions presented in this manuscript, the authors must confirm the absence of residual crystal proteins in the spore preparations used for infection by increasing Western blot exposure times. They must also directly monitor early stages of infection (spore germination and bacterial growth) particularly using the acrystalliferous Bt strains 407 and 4Q7.

Reviewer #2: None

Reviewer #3: Question 1:Cry1Ab and cry4Ba gene cloning, in all the versions, P1P2-cry1Ab, pP1P2-cry4Ba, pP4-cry1Ab, and pP4-cry4Ba, were checked by sequencing after cloning? To check that the constructions were well done.

Question 2: Could the authors estimate the amount of protein produced in vivo in larvae's midguts? Does this amount correlate with the amount of protein needed to kill the percentages of larvae shown in Figure 3? The LC values for both species are known.

**Part III – Minor Issues: Editorial and Data Presentation Modifications**

Reviewer #1: -

Reviewer #2: See Part I

Reviewer #3: Minor corrections:

93_ β-strands, instead b-strands.

99_ There is a extra “.)”

PLOS authors have the option to publish the peer review history of their article (what does this mean? ). If published, this will include your full peer review and any attached files.

**Do you want your identity to be public for this peer review?** For information about this choice, including consent withdrawal, please see our Privacy Policy .

Reviewer #1: No

Reviewer #2: No

Reviewer #3: **Yes:** Iñigo Ruiz de Escudero

**Figure resubmission:**

**Reproducibility:**



---

## [Decision Letter · Decision Letter 1]

18 Feb 2026

PPATHOGENS-D-25-02888R1

Regulatory diversity in Bacillus thuringiensis cry genes reveals flexible evolutionary strategies for in vivo toxin expression

PLOS Pathogens

Dear Dr. Bravo,

Thank you for submitting your manuscript to PLOS Pathogens. After careful consideration, we feel that it has merit but does not fully meet PLOS Pathogens's publication criteria as it currently stands. Therefore, we invite you to submit a revised version of the manuscript that addresses the points raised during the review process.

We look forward to receiving your revised manuscript.

Kind regards,

Jens Rolff

Academic Editor

PLOS Pathogens

Debra Bessen

Section Editor

PLOS Pathogens

Sumita Bhaduri-McIntosh

Editor-in-Chief

PLOS Pathogens

orcid.org/0000-0003-2946-9497

Michael Malim

Editor-in-Chief

PLOS Pathogens

orcid.org/0000-0002-7699-2064

**Additional Editor Comments:**

Thanks very much for your thorough revision and clear rebuttal. There are only a few minor issues to address now.

**Journal Requirements:**

Please amend your detailed Financial Disclosure statement. This is published with the article. It must therefore be completed in full sentences and contain the exact wording you wish to be published.

**Reviewers' Comments:**

Reviewer's Responses to Questions

**Part I - Summary**

Reviewer #2: The authors have addressed my previous comments to my satisfaction. I only have a few very minor comments to add now.

**Part II – Major Issues: Key Experiments Required for Acceptance**

Reviewer #2: None

**Part III – Minor Issues: Editorial and Data Presentation Modifications**

Reviewer #2: 1) Line 17 I suggest " while it postively regulates cry1Ac, since Spo0A mutants showed....."

2) Line 127 factor not factors

3) Line 497 delete "in the"

4) Line 501 "those originally" rather than "the originally"

5) Line 518. While it is well established that Cry2Ab (unlike Cry2Aa) has no activity against Aedes aegypti it has been reported to have some dipteran activity (10.1111/j.1574-6968.2011.02403.x )

PLOS authors have the option to publish the peer review history of their article (what does this mean? ). If published, this will include your full peer review and any attached files.

**Do you want your identity to be public for this peer review?** For information about this choice, including consent withdrawal, please see our Privacy Policy .

Reviewer #2: No

**Figure resubmission:**
---

## [Editor Report · Decision Letter 2]

19 Feb 2026

Dear Dr. Bravo,

We are pleased to inform you that your manuscript 'Regulatory diversity in Bacillus thuringiensis cry genes reveals flexible evolutionary strategies for in vivo toxin expression' has been provisionally accepted for publication in PLOS Pathogens.

Best regards,

Jens Rolff

Academic Editor

PLOS Pathogens

Debra Bessen

Section Editor

PLOS Pathogens

Sumita Bhaduri-McIntosh

Editor-in-Chief

PLOS Pathogens

orcid.org/0000-0003-2946-9497

Michael Malim

Editor-in-Chief

PLOS Pathogens

orcid.org/0000-0002-7699-2064

Thanks for your swift revision. There are no further comments.
---

## [Editor Report · Acceptance letter]

Dear Dr. Bravo,

We are delighted to inform you that your manuscript, "Regulatory diversity in Bacillus thuringiensis cry genes reveals flexible evolutionary strategies for in vivo toxin expression," has been formally accepted for publication in PLOS Pathogens.

Best regards,

Sumita Bhaduri-McIntosh

Editor-in-Chief

PLOS Pathogens

orcid.org/0000-0003-2946-9497

Michael Malim

Editor-in-Chief

PLOS Pathogens

orcid.org/0000-0002-7699-2064